# A 3-D groundwater isoscape of the contiguous USA for forensic and water resource science

Gabriel J. Bowen*, Jessica S. Guo[¤a], Scott T. Allen[¤b]

Department of Geology & Geophysics and Global Change & Sustainability Center, University of Utah, Salt Lake City, UT, United States of America

¤a Current address: College of Agriculture and Life Sciences, University of Arizona, Tucson, AZ, United States of America
¤b Current address: Department of Natural Resources and Environmental Science, University of Nevada, Reno, NV, United States of America
* gabe.bowen@utah.edu

## Abstract

A wide range of hydrological, ecological, environmental, and forensic science applications rely on predictive "isoscape" maps to provide estimates of the hydrogen or oxygen isotopic compositions of environmental water sources. Many water isoscapes have been developed, but few studies have produced isoscapes specifically representing groundwaters. None of these have represented distinct subsurface layers and isotopic variations across them. Here we compiled >6 million well completion records and >27,000 groundwater isotope data-points to develop a space- and depth-explicit water isoscape for the contiguous United States. This 3-dimensional model shows that vertical isotopic heterogeneity in the subsurface is substantial in some parts of the country and that groundwater isotope delta values often differ from those of coincident precipitation or surface water resources; many of these patterns can be explained by established hydrological and hydrogeological mechanisms. We validate the groundwater isoscape against an independent data set of tap water values and show that the model accurately predicts tap water values in communities known to use groundwater resources. This new approach represents a foundation for further developments and the resulting isoscape should provide improved predictions of water isotope values in systems where groundwater is a known or potential water source.

## Introduction

The physics of moisture evaporation, transport, and condensation in the atmosphere create coherent spatiotemporal variation in the hydrogen (H) and oxygen (O) isotope ratios of meteoric waters [1,2]. This variation is frequently preserved and passed on through the hydrological cycle and the transfer of H and O into Earth materials and ecosystems, where it is widely useful in identifying and partitioning the sources of water and geological or biological materials. For example, isotopic measurements of waters have been used to assess sources and mechanisms

**Data Availability Statement:** All water isotope data used in this study can be accessed through the Waterisotopes Database (wiDB, http://waterisotopesdb.org/). A list of datasets used here is provided as a supplementary table. A small

number of these data are not available for redistribution by the authors or through the wiDB, in these cases the database record provides information on how the data can be obtained from 3rd party sources. All codes and ancillary data used for analysis and production of figures have been archived on Zenodo (https://doi.org/10.5281/zenodo.5554984). State-level well records used in this study are considered sensitive information by some states and can not be redistributed as a product of our research. We provide a list of the data sources from which these data were obtained as a supplementary table.

**Funding:** Support for this research was provided by a grant (#4666) from the US Department of Defense and the Henry M Jackson Foundation for the Advancement of Military Medicine (https://www.hjf.org/). The funders conducted a compliance review of the manuscript prior to submission, but otherwise had no role in study design, data collection and analysis, decision to publish, or preparation of the manuscript.

**Competing interests:** The authors have declared that no competing interests exist.

of runoff generation [3,4], groundwater recharge [5,6], water use within cities [7,8], and water uptake by plants [9–11]. In animal systems, these isotopes have been widely used to reconstruct the movement and migration of animals and humans [12–14] and they may also help constrain the diets of consumers [15].

A commonality across these studies is the need to accurately characterize the isotopic composition of water sources. In small-scale, place-based research, this is often accomplished through targeted sampling and measurement campaigns. In large-scale studies, where the focus is on patterns or processes operating across regions, continents, or the globe, direct measurements are impractical or impossible. Over the past two decades, the need for source characterization in such studies has been fulfilled through the development of "isoscapes": predictive models of spatiotemporal isotopic patterns that are developed through statistical or process-based modeling, often leveraging large observational data sets for training and/or validation [16]. In order to meet the needs of a given study, an isoscape should ideally be both accurate, in terms of closely reflecting the target quantity, and specific, in the sense that the quantity represented is the one relevant to the investigation [17].

A wide range of isoscapes that capture the major synoptic patterns of water isotope variation (i.e., spatial length scales of 1,000 km and above) have been developed. Isoscapes of precipitation have a long history of development and use [18–20], but predictive isoscape models of atmospheric water vapor [21], stream water [22,23], and tap water [24–26] are also available. Despite the widespread use of H and O isotopes in the study of groundwater systems and the importance of groundwater as a source to many hydrological and drinking water systems, groundwater isoscapes have received relatively little attention. Only two studies (to our knowledge) have attempted to develop synoptic-scale groundwater isoscapes: one using spatial interpolation of groundwaters across Mexico [27] and a second [28] using machine learning methods to model shallow groundwaters of the contiguous United States (CONUS).

Although these groundwater studies made important contributions to the understanding and analysis of isotope patterns and hydrology of groundwater systems, the predictive isoscapes they developed share limitations with respect to representing two characteristics of groundwater as a water source. First, multiple aquifers may be present at different subsurface depths for a given geographic location, and water from these aquifers can have dramatically different isotopic compositions [29]. Second, groundwater at specific depths, and groundwater generally, may be inactive in the water cycle or unused by humans at a given location. Both existing groundwater isoscapes represent a single (or undifferentiated) subsurface depth interval, and both predict groundwater values continuously across their study domain. As a result, even if the resulting isoscapes offer accurate predictions, they may represent isotope ratios for groundwater resources that do not exist or are different from those contributing as sources to a given study system.

Here, we attempt to advance the development of accurate predictive groundwater isoscapes through a depth-explicit analysis of groundwater that specifically represents the presence or absence of human-exploited aquifers at different locations and depths. We start by developing a 3-dimensional (3-d) model of aquifer presence/absence using a compilation of well depth information from across the CONUS. We then compile and geostatistically model groundwater isotope data to generate a 3-d grid of isotopic predictions and validate these using an independent dataset of tap water samples from groundwater-supplied cities and towns. We discuss and interpret patterns of groundwater isotopic compositions observed at different depths, their potential hydrological causes, and their implications for applications involving isotope-based source identification of waters and Earth and ecological materials. Our analysis and discussion focuses on isotope ratios of oxygen, but equivalent analyses were conducted and data products generated for hydrogen.

## Materials and methods

Unless stated otherwise, all analyses were done in the R programming environment [version 4.0.4; 30] on a Windows desktop PC (Intel i7-8700K, 6 cores/12 processors, 32 GB RAM). The code used for data processing and production of figures are available, together with the resulting data products, in [31].

### Data

Three distinct datasets were compiled and used in this research (Table 1). Groundwater well depth (GWD) data for the 48 contiguous United States were compiled from state-level geodatabases or, where state-agency data were unavailable, from the U.S. Geological Survey (Fig 1A). Each source consisted of georeferenced, tabular data records, usually reporting information on the location, depth, use, and date of completion of groundwater wells. The precision of geographic coordinate information varied among sources, and in some cases had been degraded (e.g., to the center of a State Plane Coordinate System section) prior to reporting. In all cases the precision of reported coordinates was far greater than the spatial resolution of our analysis, however. The temporal coverage of each data source also varied, though in general the digital data we obtained was biased toward wells completed in the past several decades. Because date of well completion was not always reported we did not systematically evaluate differences in the temporal coverage of different data sources.

Initial processing for most states was done using ArcGIS 10.6, and the records were then compiled into a single spatial points data frame and stored as a Rdata object (totaling 214 MB in size). Records were screened to remove industrial, geothermal, and other non-potable wells; because the available data and metadata varied substantially among states, a custom set of screening criteria was used for each state database. We removed records lacking essential metadata (i.e., completion depth, geographic coordinates) and those with geographic coordinates that fell outside of the state in which they were reported to occur. Wells with reported depths less than 1 meter or greater than 2,000 meters, which we considered unlikely to be potable water wells in most cases, were also removed from the collection. Data sources are reported in S1 Table; because some states consider well records to be sensitive information, we are not able to publicly redistribute the full compilation.

Stable isotope ratio measurements for groundwater (GWI) were compiled from published, public domain, and unpublished sources (Fig 1B). New data were produced for 210 samples that filled identified gaps in the compiled dataset. Briefly, samples were collected directly from the well or a tap serviced by the well, stored in a sealed HDPE or glass bottle, and analyzed using a Picarro L2130-i spectroscopic analyzer at the SIRFER laboratory, University of Utah, and the methods described by Good et al. [32]. All data were registered in the Waterisotopes Database (wiDB, http://waterisotopesDB.org) and a list of data sources is provided in S2 Table. Data were screened to remove samples with missing information (e.g., geographic coordinates, well depth, and $\delta^{18}O$ values; where $\delta = (R_{sample} - R_{standard})/R_{standard}$ and $R$ is the ratio of

**Table 1. Properties of the three primary datasets used in this work.**

| Acronym | Description | Source | # Data; # Sites | Use |
|---|---|---|---|---|
| GWD | Groundwater well completion records, including completed depth | National and state-level databases | 6,169,771; 6,169,771 | Aquifer map |
| GWI | Groundwater H and/or O stable isotope data | National databases, publications, or this work | 27,738; 16,000 | Aquifer map, groundwater isoscape |
| TWI | Tap water H and/or O stable isotope data for groundwater-served sites | Publications or this work | 273; 273 | Groundwater isoscape validation |

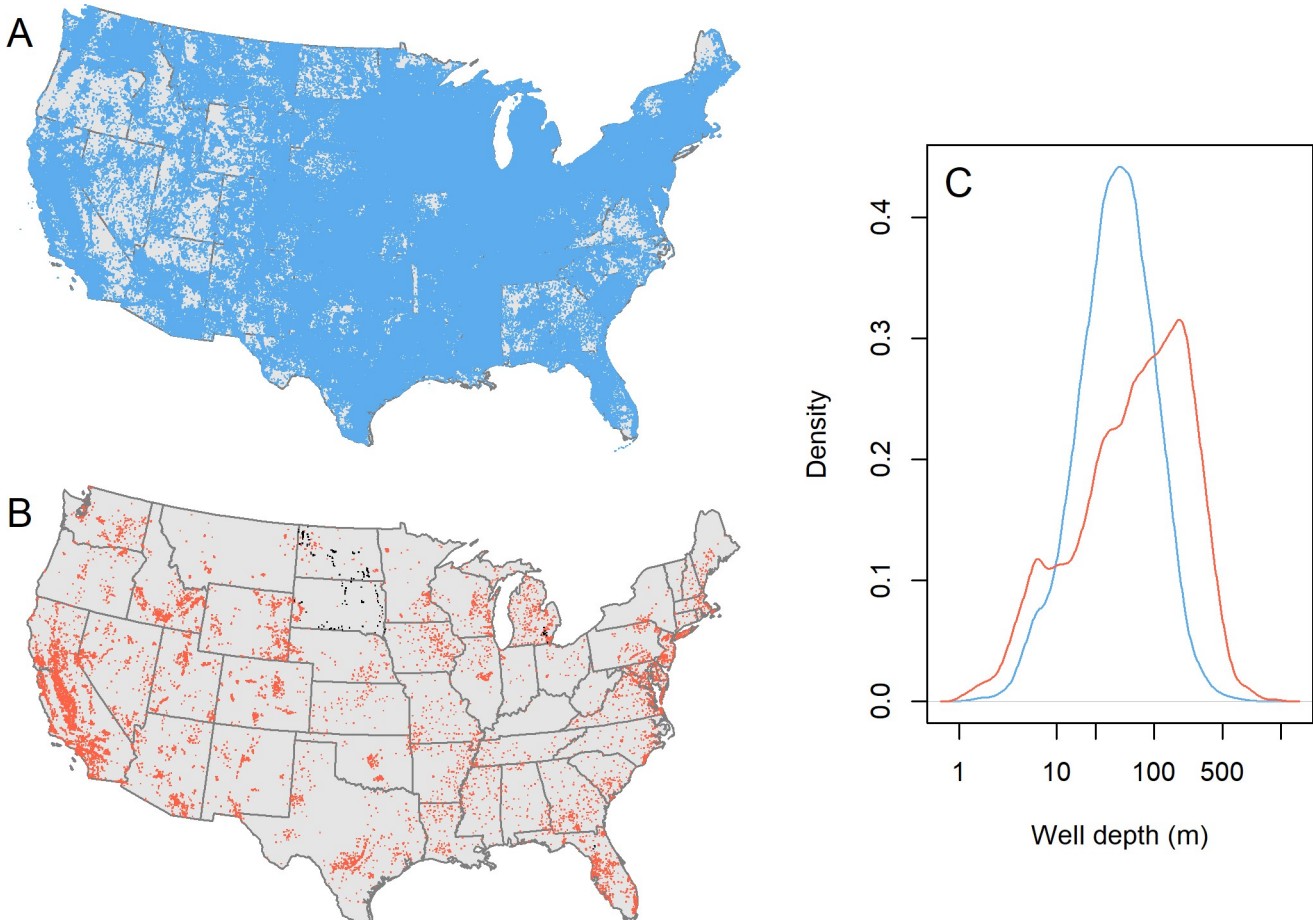

**Fig 1. Properties of the GWD and GWI datasets.** (A) Spatial distribution of GWD records. (B) Spatial distribution of GWI records. Black points represent new data produced for this study, all other data were compiled from other sources. (C) Distribution of well depths in the GWD (blue) and GWI (red) datasets.

concentrations of the heavy and light isotopes and those with deuterium excess values ($d = \delta^2H - 8 \times \delta^{18}O$) that are atypical of fresh groundwater (i.e. less than -10‰, suggesting substantial evaporation, or greater than +25‰, suggesting potential analytical error or isotopic exchange with bedrock). The isotopic data represent samples collected between 1970 and present, with relatively constant temporal sample density between 1984 and 2019 and fewer samples in the earlier and most recent parts of the sampling interval.

For validation purposes, we used isotopic data for 273 tap water samples (TWI) known to be of groundwater origin. These samples differed from the GWI samples in that they generally could not be associated directly with a single well, but they came from buildings, towns, or cities known to use groundwater based on public water records (in most cases information reported in annual Consumer Confidence Reports, CCRs) or interviews with owners or managers. This dataset included 79 sites collected in spring 2019 (collection and analysis methods as described above), and an additional 194 sites from the dataset of Bowen et al. [24].

## Aquifer map

We used the GWD and GWI databases to develop a gridded, 3-d map of aquifers (here defined as subsurface strata from which humans have extracted or are actively extracting water as a

resource) across the contiguous USA. We started with the assumption that wells were completed (bottom depth) within a productive aquifer unit. Although well completion depth is generally deeper than the subsurface depth of the aquifer layers from which the well draws, other potential metrics for the depth at which water was produced were inconsistently available (water depth, screened interval). In general, wells are completed within or just below a target aquifer, and thus we consider completion depth to be a consistent if slightly biased measure of the depth from which water is produced.

Our gridded map was produced on a regular 25 x 25 km grid (Albers Equal Area projection) using 7 depth intervals distributed between 1 and 2,000 meters (bounded by 1, 10, 25, 50, 100, 200, 500, and 2,000 m). This resolution was selected as a compromise based on practical (e.g., computational demand and data size, scaling of isotopic variance with distance) and theoretical (e.g., approximate length scale of aquifers, likely increase in mixing with depth) considerations. With respect to aquifer presence/absence, the 25 km spatial resolution is likely too coarse to represent some shallow, heterogeneous (e.g., alluvial) aquifers, but should be adequate to capture patterns of presence/absence of such aquifers and to map the general footprint of most basinal and regional aquifers. With respect to isotopic variability, the average variance among $\delta^{18}$O values from multiple wells within a voxel increases continuously with grid cell size, but remains below 0.53‰, similar to the uncertainty in the predicted mean grid cell values, for the 25 km grid (see below). Finally, our semivariogram analysis (see below) suggests that systematic spatial changes in groundwater $\delta^{18}$O values over 25 km are generally trivial (up to a few tenths of a per mil), implying this scale is adequate to resolve spatial patterns in groundwater isotopic composition.

For each voxel in the grid, we identified whether one or more wells existed at that location and depth. If so, the voxel was classified as having an aquifer present, and if not, it was classified as aquifer absent. The classification was conducted separately using the GWD and GWI databases to assess space/depth overlap between the data sets, and the final aquifer map included information from both data sets (i.e., presence of a well in either database was taken to reflect the presence of an aquifer).

## Groundwater isoscape

We conducted both 2-dimensional (2-d) and 3-d variogram analysis of the GWI data using the *gstat* package [33,34]. Because the variogram tools do not allow co-located points, isotope $\delta$ values for samples which had identical geographic coordinates and depths (for the 3-d analysis) or depth intervals (2-d) were averaged. For the 3-d analysis, the dataset was analyzed as a spatiotemporal data object (package *spacetime*; [35,36]) wherein depth was treated as the time dimension. Because the computation time required to generate 3-d semivariograms from the full dataset would have been prohibitive, we conducted and compared analyses on two randomly selected subsets of 4,000 samples (each took ~35 minutes running on 10 processors). We used the results of the variogram analysis to determine whether prediction via 3-dimensional kriging was warranted. Given that no systematic autocorrelation of isotope $\delta$ values in the depth dimension was found (see results), 3-d kriging was not supported and our subsequent analysis used independent semivariogram modeling and 2-d kriging of groundwater isotope data from each subsurface layer. Our 2-d semivariogram modeling used a Matern covariance model with a nugget parameter.

GWI values from each subsurface depth layer were extracted and used together with that layer's 2-d semivariogram model to generate block kriging estimates predicting mean groundwater $\delta^{18}$O values and prediction variance at all well-containing voxels within the layer. Predictions from the seven layers were stacked to generate a 3-d grid. For the prediction

uncertainty stack, values for each layer were calculated by propagating the block kriging prediction variance, which represents the uncertainty in the predicted mean value for that voxel, and the average within-voxel variance for that depth interval, representing the expected distribution of individual well values around the mean. Our propagation assumes independence of these two sources of uncertainty, which we believe is appropriate given that they represent two distinct error-generating processes, but to the degree that this assumption is incorrect the uncertainty estimates will be conservative. We conducted leave-one-out cross-validation for each layer to approximate the distribution of prediction errors. Finally, the GWI data and each groundwater isoscape layer was compared with a geostatistical prediction map of mean annual modern precipitation [climatological, based on monitoring data from 1960—present; 37,38] to create maps of the difference between local groundwater and precipitation isotope predictions. The *cubeview* R-package [39] was used to export visualizations of all 3-d well presence and isotopic data products.

## Validation

We assessed the groundwater isoscape against the independent TWI dataset from cities and towns known to be serviced exclusively by groundwater. Specific information on the aquifers and well depths of the wells used by each community were not compiled, and our analysis focused on metrics that evaluated whether the distribution of source water values mapped at the geographic location of the cities or towns approximated the observed TWI values. Although it is possible that water used at some of these sites may have been transported laterally (i.e., across grid cell boundaries) from its location of extraction, all source-water isoscapes used here are spatially smoothed and it is unlikely that lateral transport would lead to large or systematic inaccuracies in our validation analysis. We produced a set of 2-d summary layers from the 3-d groundwater isoscape (including the mean and standard deviation of values from the subsurface layers represented at each geographic grid cell). Model fit was assessed by regressing observed tap water $\delta^{18}O$ values against associated depth-averaged means from the groundwater isoscape; an equivalent comparison was made with values from the precipitation and Stahl et al. [28] CONUS groundwater isoscapes. For each TWI site, we also evaluated whether the measured tap water value was contained within the mean ± $z$ standard deviation (σ) groundwater $\delta^{18}O$ prediction interval for each subsurface layer (where $z$ was 1 or 1.96 to approximate 68% and 95% prediction intervals, respectively).

## Results and discussion

### Aquifer map

The GWD database consists of 6,169,771 records. The number of records per state varies widely and is much lower for the states without state-agency data (e.g., minimum = 2,728, Rhode Island) than for those with (e.g., maximum = 397,627, Indiana). The GWI database consists of 27,738 records from the 48 contiguous states (minimum = 3, Rhode Island; maximum = 9,707, California) with $\delta^{18}O$ values ranging from –21.7 to +4.7‰. These records come from at least 16,000 unique wells, based on the number of records with distinct geographic coordinates. Depth distributions for the GWD and GWI databases are comparable to first order, but in contrast to the log-normally distributed GWD depths the GWI samples overrepresent shallower and deeper aquifers and underrepresent intermediate-depth wells (Fig 1C). This likely reflects targeted sampling of large, deep aquifers in isotope hydrology monitoring and research studies.

　　The composite map represents aquifer presence and absence in 87,969 voxels across the CONUS (Figs 2 and S1). One or more wells are present within 68.7% of these cells. Well-

containing voxel density varied substantially with depth. Within the deepest (500–2,000 m) layer 19% of grid cells contained wells, whereas 89% of cells within the 50–100 m layer contained wells; well density varied continuously between these extremes. Visual review of the mapped distribution suggests that coverage is fairly even across most states, with the notable exception of shallow subsurface depths (particularly less than 25 m) within states for which no state-level GWD data were available (e.g., Georgia, Alabama, Virginia, West Virginia) and the state of Missouri. In the former case, we suggest that the well records obtainable from the USGS database are not representative of the distribution of shallow well depths within these states, being biased toward monitoring of deeper aquifer systems. We were not able to identify a clear explanation for the lack of shallow wells in the Missouri state data set; the distribution of depths for this state includes few wells shallower than ~10 m but is otherwise similar to those for other nearby and hydrogeologically-related states. The GWI and GWD databases also show substantial congruence. For example, 93.5% of all GWI data come from wells that occur within voxels identified as aquifer-bearing based on the GWD data, and a large number of the remaining GWI data occur within states lacking state-level GWD coverage.

The gridded aquifer map is an imperfect approximation because of inaccuracies inevitable in such a large compilation of heterogeneous data and known (and potentially unknown) data gaps. Regardless, the product demonstrates numerous patterns that match known hydrogeological features, suggesting that the map offers a reasonable first-order approximation of the availability and use of groundwater across well-sampled parts of the CONUS. Very shallow wells (<10 m) occur sporadically throughout the wetter parts of the country, including the upper Midwest and northern Mountain Interior, but are uncommon in many dry areas, like the Great Basin and Western Great Plains, where water tables are frequently below 10 m. Across the western Great Plains, for example, depth to water within the High Plains aquifer system is typically greater than 10 meters except near the Platte River, which is apparent in our aquifer map as a band of well-containing, shallow-layer cells (Fig 2A).

Wells of intermediate depth (10–100 m) are more ubiquitous in the dataset and thus challenging to match with distinct hydrogeological features. They are much more common across arid and semiarid regions than are shallow wells, and major aquifers known to produce within these depth intervals generally coincide with a high density of well-containing cells (Fig 2B). The High Plains and Upper Cretaceous aquifers of the Great Plains and the Willamette Valley aquifers of Oregon provide examples of the latter and are notable in that intermediate-depth wells become scarcer immediately outside of the mapped aquifer boundaries (Fig 2B).

The number of aquifer-containing cells are fewer, and their distribution patchier, at depths below 100 meters, and in many cases clusters of well-containing cells can be matched with specific aquifer systems known to produce from these depths (Fig 2C). Within the region underlain by the High Plains aquifer system, cells mapped as containing deep wells are largely limited to parts of southwest Kansas and the Texas/Oklahoma panhandle region, where long-term exploitation of groundwater has lowered water tables and led to deep drilling [41]. Elsewhere, major deep aquifer systems in the Texas coastal plains, Ozark Plateau, Denver Basin, Snake River Plain, and interior valleys of the Pacific Coast states are clearly represented in the gridded map.

## Groundwater isoscape

Both 2-d and 3-d sample semivariograms show very strong spatial autocorrelation in the horizontal plane (S2 and S3 Figs). The 2-d semivariogram model parameters suggest that at most depths this correlation structure extends to ~1,000–2,000 km, or approximately 20–40% of the length-scale of the CONUS (Table 2). The length-scale of correlation decreases with depth

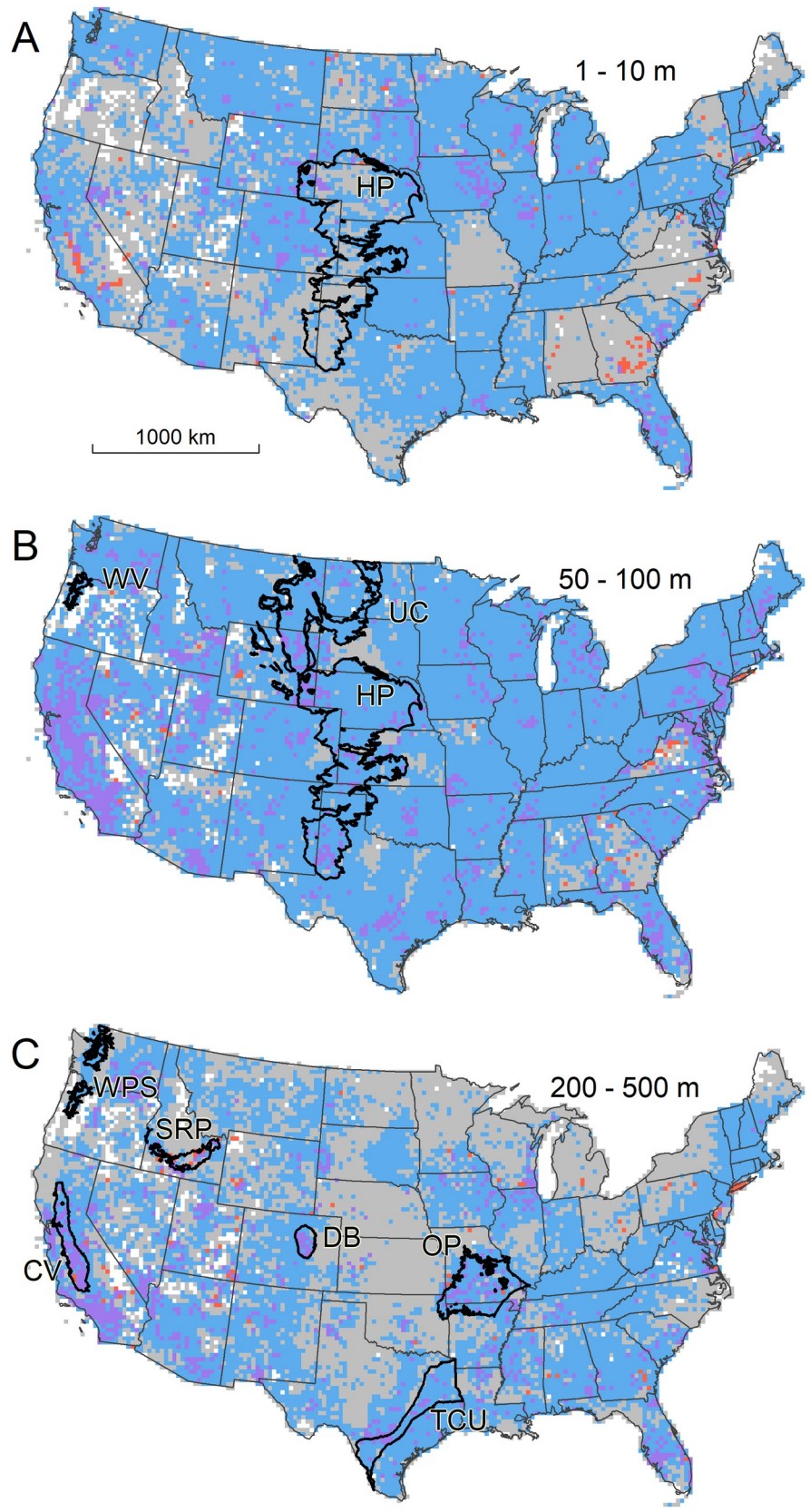

**Fig 2.** Presence or absence of wells completed in a shallow (A), intermediate (B), and deep (C) subsurface layer of the 3-D aquifer map. In all panels white cells contain no wells at any depth in either the GWD or GWI database, grey cells contain no wells at the layer's depth in either database, and blue, red, and purple cells contain one or more wells at the layer's depth in the GWD, GWI, or both databases (respectively). Black polygons show the spatial extent of major aquifers discussed in text [40]. HP = High Plains; WV = Willamette Valley; UC = Upper Cretaceous; WPS = Willamette and Puget Sound; SRP = Snake River Plain basalts; CV = Central Valley; TCU = Texas Coastal Uplands; OP = Ozark Plateau; DB = Denver Basin.

(above the deepest layer), but this variation is also strongly correlated with, and may largely be controlled by, differences in sampling density at different depths. The observed scale of correlation is similar to that of precipitation isotope ratios [20,42], which implies that geographic differences in the isotope ratios of precipitation recharging the aquifers could be a first-order control on the groundwater isotope ratios. Differences in the size and connectivity of aquifers, as well as heterogeneity in recharge age, may be secondary factors that also influence the length scale and structure of correlation within depth layers as well as the differences in semivariogram model parameters among layers. Little to no autocorrelation is apparent, regardless of horizontal lag distance, in the vertical direction (S3 Fig). The lack of a systematic increase in isotopic dissimilarity with depth may simply reflect the lack of a systematic difference in the isotopic composition of recharge to shallower and deeper aquifers due to heterogeneity in the age and conditions under which shallow and deep recharge occurred in different parts of the CONUS [e.g., 43].

Variation among multiple GWI $\delta^{18}$O measurements collocated within individual voxels is relatively low and decreases somewhat with depth (from an average standard deviation of 0.65‰ in the 0–10 m layer to 0.36‰ in the 500–2,000 m layer; mean across layers = 0.51‰). The greater sub-grid-scale homogeneity of the deep aquifer samples could reflect fundamental physical properties of the subsurface (e.g., longer residence times and greater mixing of waters in deeper aquifers) but could also reflect selective drilling of larger, more homogeneous aquifers at depth. Cross-validation prediction errors are similarly distributed for all depth intervals and are somewhat heavy-tailed (S4 Fig). Mean absolute errors and root mean squared errors range from 0.51 and 0.80‰ for the 200–500 m interval to 0.80 and 1.25‰ in the 500–2,000 m layer; for all depth intervals excluding this deepest one, the values are less than or equal to 0.71 and 1.07‰. These error statistics are very similar to those reported for geostatistically-based predictions of CONUS stream water [23] and global precipitation $\delta^{18}$O values [37] and for a CONUS shallow groundwater isoscape produced using a random forest algorithm [28].

The 1ˢᵗ-order pattern of groundwater $\delta^{18}$O variation is similar across all depth intervals analyzed here, and is characterized by low values, approaching -20‰, across the northern Rocky Mountain states, and the highest values, near or slightly above 0‰, in Florida and along the Gulf Coast (Figs 3 and S5). This pattern, as well as the measured GWI values themselves (Fig 4), closely parallels variation in precipitation $\delta^{18}$O values across the CONUS [23,44], and

**Table 2. Optimized 2-d semivariogram model parameters and number of observations (unique locations) for groundwater $\delta^{18}$O values from seven depth intervals.**

| Interval (m) | Nugget | Partial Sill | Range (km) | Kappa | Observations |
|---|---|---|---|---|---|
| 0–10 | 1.3 | 61.7 | 5,383 | 0.5 | 1,954 |
| 10–25 | 1.1 | 62.5 | 4,177 | 0.5 | 2,130 |
| 25–50 | 0.2 | 34.7 | 2,022 | 0.4 | 2,825 |
| 50–100 | 0 | 28.5 | 1,308 | 0.4 | 3,609 |
| 100–250 | 0.2 | 26.7 | 1,019 | 0.5 | 3,642 |
| 250–500 | 0.7 | 31.8 | 1,303 | 0.6 | 2,526 |
| 500–1,000 | 0.7 | 245.8 | 9,886 | 0.6 | 363 |

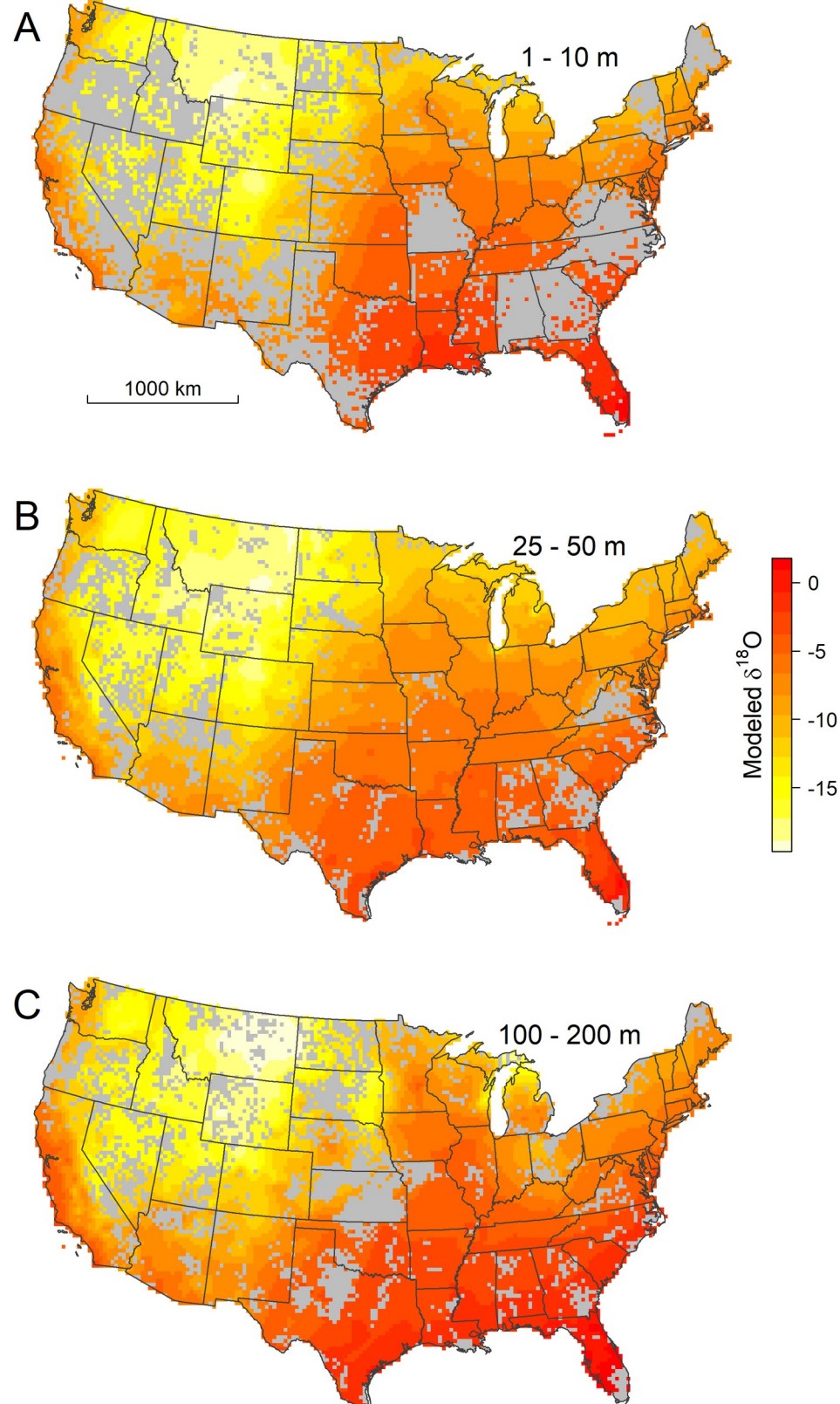

**Fig 3.** Spatially interpolated δ¹⁸O values for ground water at shallow (A), intermediate (B), and deep (C) depths. Gray cells contain no wells completed within the mapped depth interval.

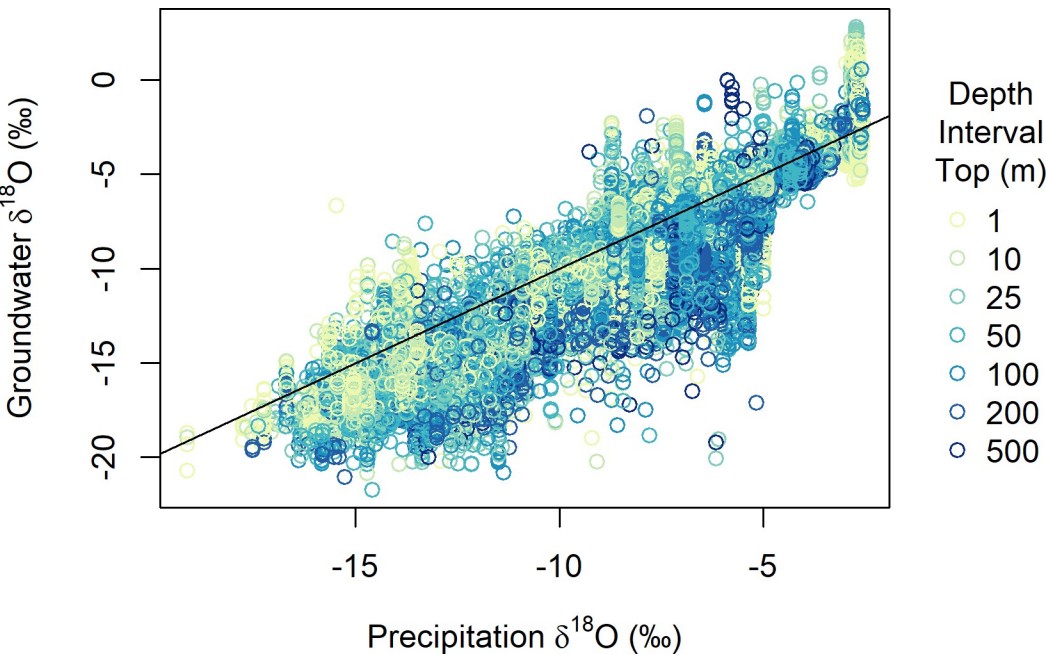

**Fig 4. δ^18O values for GWI database samples and interpolated modern, annual-average precipitation, as a function of well depth.** The 1:1 line is shown in black.

is similar to that estimated in a previous analysis of shallow (<45 m) ground waters based on many of the same data used here [28]. As noted for shallow ground water in that study, the general similarity of groundwater and precipitation patterns indicates that isotope effects associated with atmospheric water cycling are a strong determinant of groundwater isotope ratios. Our work shows that this similarity extends deep into the subsurface, which implies relative stability of both climate-driven precipitation isotope patterns and groundwater recharge processes over time, given the likely wide (but here unconstrained) range of recharge ages for waters in different regions and depths.

Quantitative comparison of the groundwater data and model with modern precipitation δ^18O estimates, however, reveals systematic regional deviations that, in many cases, exceed the local isoscape prediction uncertainty and vary with depth (Figs 4, 5, S6, and S7). Across most of the eastern USA and the southern Great Plains, groundwater δ^18O values at most depths are within ~2‰ of the annual mean precipitation values, with localized exceptions at shallow depths in south Florida (with groundwater values higher than precipitation) and across a range of depths in areas adjacent to the Great Lakes (with groundwater values lower than precipitation). We find widespread ^18O-depletion (relative to precipitation) in groundwaters across the northern Great Plains and Western Interior, as seen previously in shallow groundwater [28], streamflow [23,45], and tap water [24]. The pattern recovered here, however, shows substantial variation with depth, with larger-magnitude ^18O-depletion often occurring in deeper groundwaters. The tendency for deeper groundwaters and groundwaters from higher-latitude and continental interior sites to be more ^18O-depleted relative to estimated modern, local precipitation is also expressed in the raw GWI data (Fig 4). The average groundwater-precipitation offset decreases from -0.8‰ for depths < 50 m to -2.0‰ for wells deeper than 200 m, and groundwater δ^18O values substantially (e.g., 5‰) lighter than local precipitation are not found at any depth at low-latitude and coastal sites where mean annual precipitation δ^18O values exceed -5‰. Groundwater that is ^18O-enriched relative to precipitation

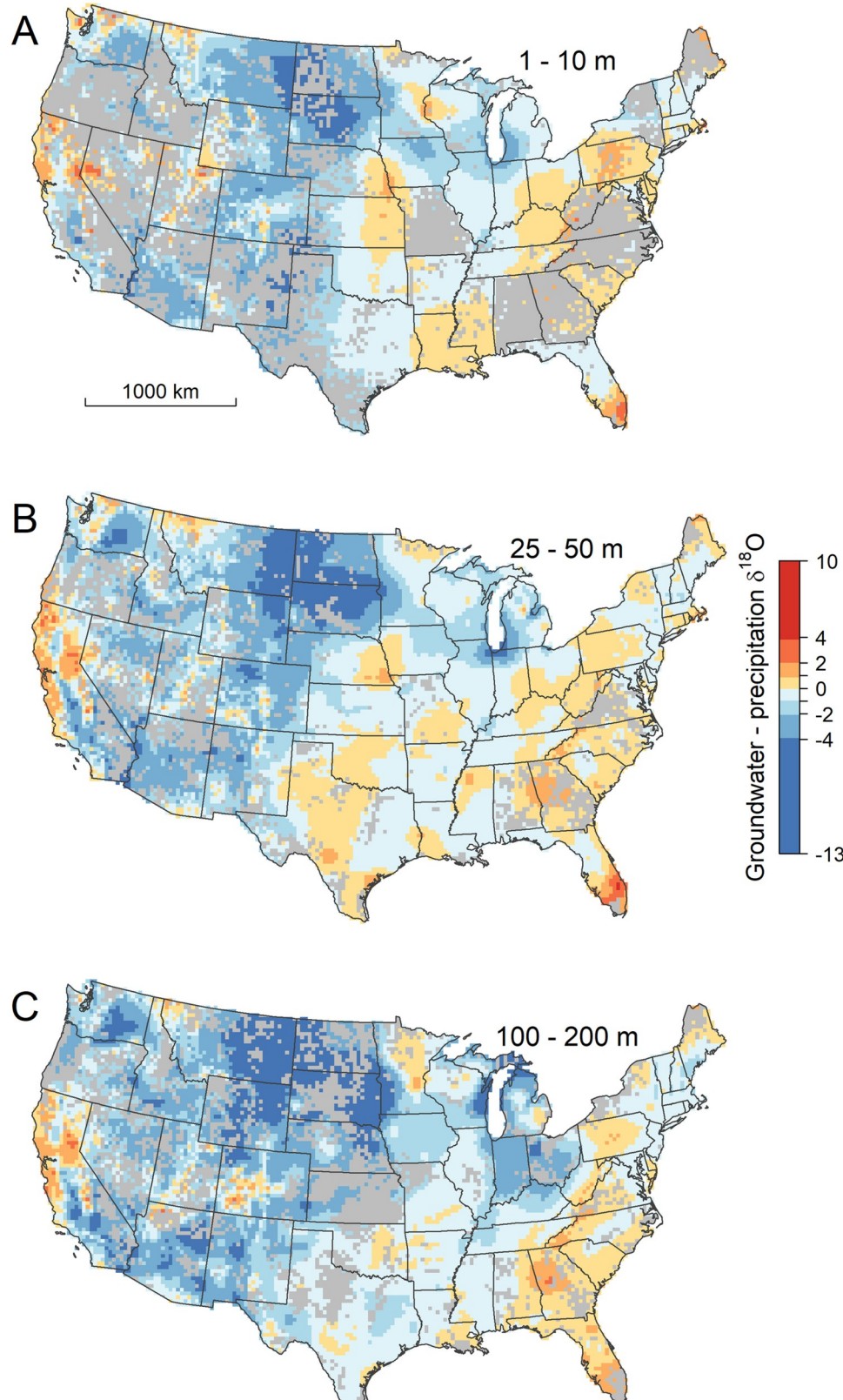

**Fig 5.** The difference in $\delta^{18}O$ values between interpolated groundwater and modern, annual-average precipitation at shallow (A), intermediate (B), and deep (C) depths. Gray cells contain no wells completed within the mapped depth interval.

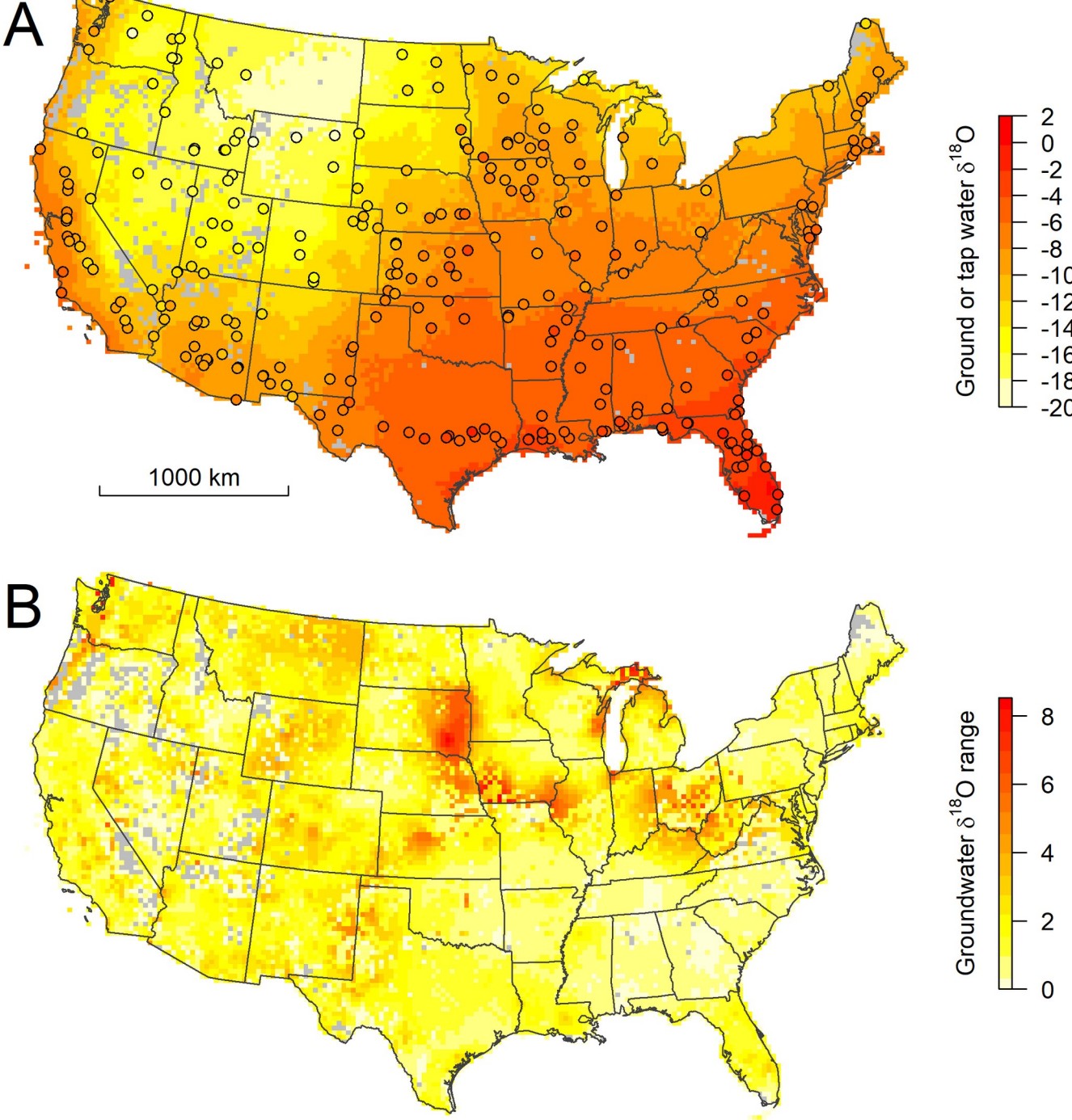

**Fig 6.** 2-d summary statistics (A: Mean, B: Standard deviation) for 3-d groundwater isoscape. Points in panel A show measured tap water δ values for validation sites. Gray cells contain no wells completed at any depth interval.

occurs in regional zones across parts of Colorado, Utah, Idaho, and western Montana and at all depths throughout much of central and northern California (Fig 5).

We hypothesize that four factors could collectively explain most of the observed groundwater-precipitation isotopic differences observed here. First, much recent work has demonstrated

that differences in recharge or runoff ratios between seasons are expressed as and can be deduced from isotopic offsets between ground or surface waters and amount-weighted mean-annual precipitation [6,28,46,47]. Stahl et al. [28] present a quantitative analysis of this mechanism for their shallow groundwater data set, and suggest that preferential infiltration of cool-season precipitation across much of the western USA and warm-season precipitation in parts of the Great Plains and Atlantic Coast might explain the large-scale pattern of groundwater-precipitation offsets.

Second, many groundwaters may be derived from laterally transported, non-local precipitation, including water infiltrating from losing reaches of rivers [48], structurally-controlled recharge in mountainous regions [49], water in large or connected aquifers conducive to lateral flow [50], and water naturally or artificially recharged following inter- or intra-basinal transfer by humans [5]. In most cases, this process is expected to result in recharge of higher-elevation water with lower-than-local $\delta^{18}O$ values, and this may be a dominant process in mountainous regions of the western CONUS, along the Colorado River corridor, and in parts of the southwest where artificial recharge is widely used for storage and aquifer replenishment. Lateral surface transport may also be accompanied by evaporative loss, particularly in warm, arid regions and where transport and storage times are extended, and in some cases this may produce heavy-isotope evapoconcentrated recharge with higher $\delta^{18}O$ values than unevaporated, locally-sourced water.

Third, paleo-groundwaters may have isotope ratios different from modern precipitation solely because of climate-driven changes in precipitation (and thus recharge) isotope δ values over time. This mechanism has been invoked to explain lower-than-modern values for late glacial-age water in aquifers of the Great Basin [51] and upper Midwest [52]. Although the relationship between groundwater age and depth is indirect, in general we expect deeper waters to be older. Thus the increasing magnitude of negative groundwater-precipitation $\delta^{18}O$ offset with depth across much of the Western Interior and northern Great Plains might suggest a role for lighter, pre-Holocene recharge in driving the pattern of offsets at depth. In mountainous regions, this depth pattern is also consistent with models for structurally-controlled mountain-block recharge, and thus with the second mechanism described above, in that these models invoke routing of isotopically-lighter, higher-elevation recharge to deeper basinal aquifers [53]. Additional support for the hypothesized role pre-Holocene recharge plays in driving isotopic variability in the subsurface comes from the geographic pattern of between-layer isotopic variation in our 3-d isoscape (Fig 6B), which shows a band of high isotopic variability, driven by anomalously light groundwater isotope δ values below 100 m depth, from South Dakota to Ohio and West Virginia. This band approximately parallels the maximum extent of the Laramide ice sheet and encompasses an area throughout which end-glacial-age groundwaters occur widely [54] and likely contribute to subsurface isotopic heterogeneity.

A fourth factor that might contribute to the observed groundwater-precipitation isotopic offsets is error in the spatial models used in the analysis, in particular the precipitation model, which is based on a relatively sparse data set [38]. Failure of the precipitation isoscape to adequately represent coast-to-interior gradients along the Pacific Coast, for example, has previously been invoked to explain positive tap water-precipitation offsets observed in a similar analysis [24]. In their assessment of shallow ground waters, Stahl et al. [28] suggest that positive offsets in this region may reflect recharge by isotopically heavy fog drip-water. Although this is a plausible mechanism for some parts of the region, it is less plausible for strong positive offsets observed in and east of the northern Sierra Nevada Mountains. Our analysis also suggests that similar-magnitude offsets persist within deep aquifers of this region, which could reflect a consistent set of recharge-based controls across a wide range of depths or, perhaps more parsimoniously, can be explained if the precipitation model against which all

groundwater layers were compared is biased toward low $\delta^{18}$O values. Replication of the groundwater-precipitation comparison using another recently published precipitation isoscape [55] yielded very similar offset patterns, suggesting that data limitations, more than interpolation methodology, might limit the accuracy of these products if and where errors in the precipitation isoscapes are responsible for observed groundwater-precipitation offsets.

This analysis implies that large-scale patterns of groundwater isotope ratios in both space and depth preserve useful information on regional mechanisms of recharge. Subsequent work that refines and focuses our understanding of the relative importance of these different mechanisms is warranted. Regardless of the underlying drivers, however, the results here show that substantial differences between groundwater and meteoric water isotope ratios, and between groundwater values at different depths, exist across many parts of the CONUS. This implies that models representing the depth-specific isotopic composition of groundwater are an important, and previously missing, resource for isotope-based hydrological and forensic applications in systems where groundwater is a potential source of water, hydrogen, or oxygen.

## Validation

The geographic distribution of measured TWI values for groundwater-served communities closely matches that of the mean groundwater predictions across the CONUS (Fig 6A). The similarity of the patterns in the two datasets is also apparent when extracted average groundwater predictions are plotted against the tap water observations: the relationship between the two measures has a slope of 1 and the groundwater model predicts 92% of the variance in the tap water dataset (Fig 7A). The same comparison for the shallow groundwater $\delta^{18}$O values predicted by Stahl et al. [28] also shows that their predictions are unbiased, but has a slightly lower explained variance (89%, Fig 7B), suggesting that inclusion of information on deeper groundwater values may be important for representing isotope ratios of groundwater resources used by some communities. These relationships contrast with a comparison between tap water and modeled mean-annual precipitation $\delta^{18}$O values, which explains 83% of the tap water variance but exhibits substantial bias, particularly for sites with low $\delta^{18}$O values (Fig 7C). A similar comparison between tap water values and modeled values for nearby surface water sources was conducted by Bowen et al. [23], and also showed a reduction in bias when stream water (instead of precipitation) was used as a proxy for tap water; the degree of improvement was smaller, but that study also made no attempt to analyze only surface-water-supplied cities and towns. It seems clear, then, that the common processes driving differences between the surface- and ground-water resource isoscapes and meteoric precipitation underlie a substantial part of the previously-noted local offsets between tap water and precipitation across the CONUS [24]. Thus, models focused on drinking-water resources may provide a foundation for a more mechanistic and/or dynamic approach to studying the isotopic composition of water consumed by humans, including temporal variation associated with changes in water use and infrastructure.

A unique feature of the groundwater isoscape relative to other such data resources is that it represents both presence/absence and predicted isotope $\delta$ values of multiple sources of subsurface water at each location within the model domain. Thus, the 2-d parametric summarization of mean and variance across layers used in the preceding paragraph both simplifies and potentially misrepresents the information contained in the full 3-d product, which is comprised of multiple independent distributions. To provide a second check on the groundwater predictions that better reflected the information contained in the 3-d isoscape, we tested whether each TWI sample $\delta$ value was contained within the 68% (1$\sigma$) or 95% (1.96$\sigma$) prediction interval for any of the underlying groundwater layers. Almost 91% and 99% of the tap water values

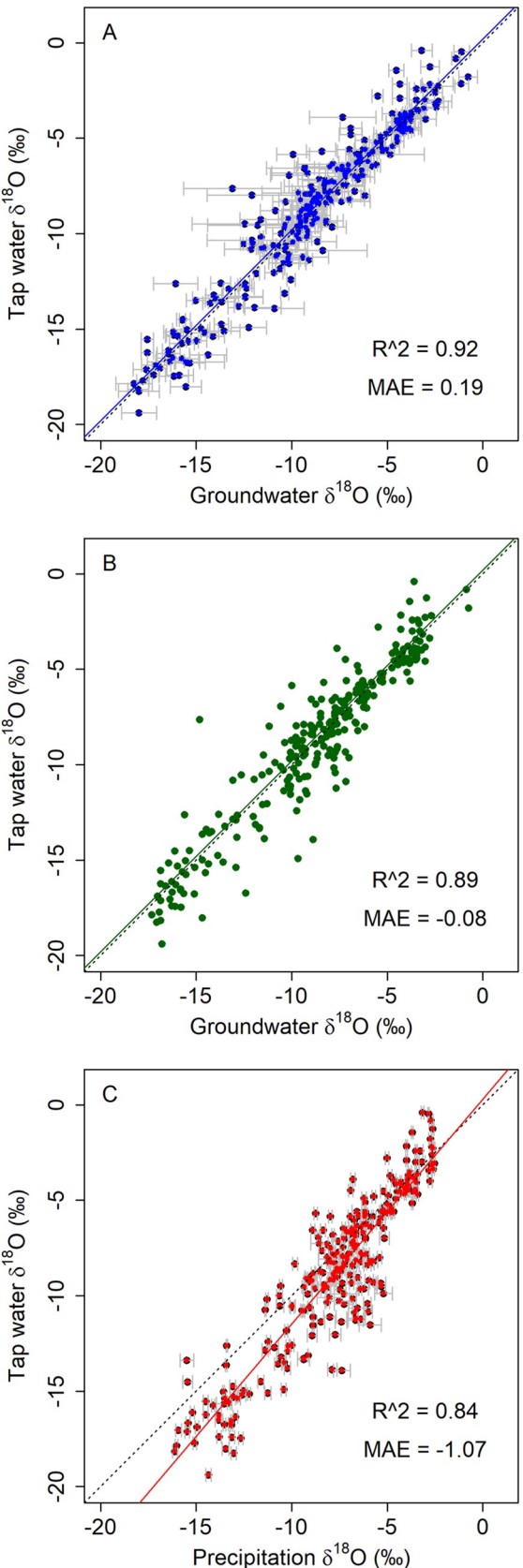

**Fig 7.** Validation plots showing comparing modeled groundwater from this study (A) or Stahl et al. [28; B] and precipitation (C) δ[18]O values with TWI values at 273 groundwater-served cities and towns. Error bars represent one standard deviation of values for different groundwater layers in A, and one standard deviation prediction uncertainties in C; equivalent location-specific uncertainty estimates were not available for the groundwater model used in B. MAE = mean absolute error.

matched the groundwater predictions at the 68% and 95% levels, respectively. Although this result may suggest that our method over-estimates prediction uncertainty, the result is likely affected by type II errors (false positives) in which the tap water samples coincidentally fall within the prediction interval for an aquifer layer from which they are not derived. A more rigorous test would require specific information on the depth of source(s) used by each town, and in the absence of this our test demonstrates that, at minimum, the depth-explicit isoscape predictions provide a conservative estimate of the range of possible groundwater source values at sites across the country. In contrast, only 7% and 14% of the samples matched local precipitation predictions within the same prediction intervals, emphasizing the importance of using water-resource-specific data products to represent the isotope ratios of non-precipitation water sources in hydrology and provenance research.

We were not able to conduct a fully equivalent test of the shallow groundwater isoscape because the Stahl et al. (2020) paper provided spatially-explicit uncertainty estimates only in figure form. We thus used the validation root mean squared error (RMSE; 1.17‰) reported by the authors to represent uncertainty in their isoscapes at all grid cells; we note that this value is ~6 times higher than the average prediction uncertainty reported for the random forest model, but we considered it more likely to be an accurate representation of the model's power to predict out-of-sample observations. Given this approximation, 70% and 90% of the tap water sample values fell within the 68% and 95% prediction intervals, respectively. This suggests that the shallow groundwater isoscape, like our 3-d product, better represents the isotope ratios of water sources used by the sampled cities and towns than does the precipitation model. At the 95% level, however, the shallow isoscape fails to predict the observed TWI values at approximately twice as many locations as would be expected; at all but one of these sites the 3-d isoscape does predict the observed values. Sites where the shallow groundwater isoscape does not predict the observed tap water values tend to be locations where the 3-d model suggests a wide range of subsurface isotopic variation, with average ranges of 1.7‰ for all tap water sites and 2.1‰ and 2.7‰ for sites where the tap δ values fall outside of the shallow isoscape's 68% and 95% prediction intervals, respectively. Spot comparisons show examples where this difference is clearly related to the use of deeper groundwater resources that are not represented in the shallow groundwater isoscape. Both Gallup, NM, and Monmouth, IL, for example, draw water primarily from deep (>100 m) wells tapping confined aquifers (based on CCRs and other public data). Our 3-d isoscape and the shallow groundwater isoscape of Stahl et al. (2020) give similar predictions for shallow groundwater at these locations, but the 3-d analysis predicts substantially lower δ[18]O values for deeper aquifers that are consistent with the measured TWI sample values. Collectively, these observations suggest that while single-layer groundwater isoscapes may offer reasonable estimates of the isotope ratios of aquifer-derived water in many cases, representation of vertical heterogeneity in the subsurface can further improve predictions and decrease the potential for mis-representation of source water values.

## Conclusion

In a wide range of large-scale hydrological and geographic provenance studies, there is a need for isoscapes that provide accurate predictions of the specific water resources of relevance to the study system. Despite the availability of isoscapes suitable for many systems, development

of large-scale groundwater isoscapes has been limited. Moreover, those data products which have been created have adopted a continuous, 2-d model for the subsurface that does not reflect the existence of multiple, discrete hydrogeological units and spatial variations in the presence/absence of different units (and active groundwater in general). Our work addresses this issue through the development of a new space- and depth-explicit groundwater isoscape from the CONUS, which shows strong performance in predicting isotope ratios of tap water sampled in cities and towns known to use groundwater resources. We suggest that the 3-d aquifer map presence/absence map may be useful in large-scale hydrology and water resource studies that require a spatially extensive (though low-resolution) model of aquifer distributions and/or models of what groundwater resources are actively exploited by humans. The 3-d groundwater isoscape represents an important step toward making groundwater-specific isotopic predictions available for large-scale hydrological, ecological, environmental, and forensic research.

Despite this positive advance, several limitations still exist with respect to the further development and use of groundwater isoscapes. First, the approach used here still simplifies both the geometry and the spatial covariance structure of groundwater isotope ratios, and opportunities exist to further improve on these aspects of this work. Modeling approaches that identify and model 3-d covariance structures that respect discrete flow boundaries between aquifers— e.g., confining layers whose depth may change over space—would improve hydrogeological accuracy over the continuous and isotropic approach used here. Second, our methods draw their power from large empirical datasets, and are unlikely to perform as well in locations where extensive well records and groundwater isotope data are unavailable. Even within the CONUS, we found that lack of publicly available well records limited our ability to create a realistic model for some states, such as Georgia and Missouri. Here, modeling approaches that incorporate more information on physiochemical processes, either through first principles or extrapolation of process-driven empirical relationships calibrated in data-rich regions, may produce better results. Third, the isoscape is based on data (both well records and isotopic samples) that are aggregated over time and may be heterogeneously sampled in time. The product is thus best considered a climatological representation of recent water isotope patterns, but with the caveat that the results may be biased toward different parts of the observational period in different locations and may not reflect changes in dynamic groundwater systems, especially in very shallow aquifers and those which are being unsustainably extracted. Finally, the availability of more specific isoscapes representing multiple water resources (e.g., multiple groundwater layers, surface water) raises a new issue in that studies intending to use these data products must identify which resources are relevant to the study system at different geographic locations. For studies focused on human systems, where evolving infrastructure may tap different and evolving sources over time, this may be a significant challenge. Conversely, however, this situation presents new and exciting opportunities to use source-specific isoscapes in combination with tap water isotope data to better reconstruct, understand, or monitor water resource use by people over space and time [7,56].

## Supporting information

**S1 Fig. Cubeview image of aquifer presence/absence model and data distribution.**
White = no wells at any depth; Grey = no wells at current depth; Blue = only well completion database wells at current depth; Red = only isotope database wells at current depth; Purple = wells from both databases at current depth. Z index represents the subsurface depth layer, where: 1 = 500–2,000 m; 2 = 200–500 m; 3 = 100–200 m; 4 = 50–100 m; 5 = 25–50 m; 6 = 10–25 m; 7 = 1–10 m. Views can be loaded as a widget in most web browsers, and users

can move through the X/Y/Z slices using the arrow and page up/down keys. The 3-d model can be rotated by clicking and dragging the view frame.
(HTML)

**S2 Fig. Representative 2-d semivariograms and optimized semivariogram models for two subsurface layers.**
(PNG)

**S3 Fig. Representative 3-d semivariogram map for a subset of 4,000 well isotope data.**
(PNG)

**S4 Fig. Density plot showing the distribution of Kriging cross-validation errors for each subsurface layer.**
(PNG)

**S5 Fig. Cubeview image showing mean predicted groundwater $\delta^{18}O$ values.** Z index values as in S1 Fig.
(HTML)

**S6 Fig. Cubeview image showing prediction uncertainty (1σ) for groundwater $\delta^{18}O$ values.** Z index values as in S1 Fig.
(HTML)

**S7 Fig. Cubeview image showing the difference between mean predicted groundwater and precipitation $\delta^{18}O$ values.** Z index values as in S1 Fig.
(HTML)

**S1 Table. List of data sources compiled in the GWD data set.**
(CSV)

**S2 Table. List of data sources compiled in the GWI data set, including project information the data deposited in the Waterisotopes Database (http://waterisotopesDB.org).**
(CSV)

## Acknowledgments

We thank the University of Utah DIGIT lab for assistance compiling state-level well records, the Center for High Performance Computing for data storage and computational infrastructure, the South Dakota Geological Survey and North Dakota Department of Environmental Quality for assistance with sample collection, and all researchers who shared or openly published the groundwater data used in this study. All code used to conduct the analyses and prepare the figures, along with the 3-d isoscapes produced here, are archived on Zenodo [31].

## Author Contributions

**Conceptualization:** Gabriel J. Bowen.

**Data curation:** Gabriel J. Bowen, Jessica S. Guo, Scott T. Allen.

**Formal analysis:** Gabriel J. Bowen, Jessica S. Guo, Scott T. Allen.

**Funding acquisition:** Gabriel J. Bowen.

**Investigation:** Gabriel J. Bowen.

**Methodology:** Gabriel J. Bowen, Jessica S. Guo, Scott T. Allen.

**Project administration:** Gabriel J. Bowen.

**Software:** Gabriel J. Bowen.

**Writing – original draft:** Gabriel J. Bowen.

**Writing – review & editing:** Jessica S. Guo, Scott T. Allen.

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
