## [Decision Letter · Decision Letter 0]

24 Aug 2021

PONE-D-21-21283

A 3-D groundwater isoscape of the contiguous USA for forensic and water resource science

PLOS ONE

Dear Dr. Brown,

Thank you for submitting your manuscript to PLOS ONE. After careful consideration, we feel that it has merit but does not fully meet PLOS ONE’s publication criteria as it currently stands. Therefore, we invite you to submit a revised version of the manuscript that addresses the points raised during the review process.

Please submit your revised manuscript by Oct 08 2021 11:59PM. If you will need more time than this to complete your revisions, please reply to this message or contact the journal office at plosone@plos.org. Please include the following items when submitting your revised manuscript:

We look forward to receiving your revised manuscript.

Kind regards,

Jehangir H. Bhadha, Ph.D.

Academic Editor

PLOS ONE

2. We note that Figures 2, 3, 4 and 5 in your submission contain map images which may be copyrighted. All PLOS content is published under the Creative Commons Attribution License (CC BY 4.0), which means that the manuscript, images, and Supporting Information files will be freely available online, and any third party is permitted to access, download, copy, distribute, and use these materials in any way, even commercially, with proper attribution. For these reasons, we cannot publish previously copyrighted maps or satellite images created using proprietary data, such as Google software (Google Maps, Street View, and Earth). For more information, see our copyright guidelines: http://journals.plos.org/plosone/s/licenses-and-copyright.

Additional Editor Comments (if provided):

Dear Authors. In lieu of the comments provided by 3 reviewers, I recommend the paper has potential for publication in PLOS ONE with moderate revisions. In particular, I recommend the authors pay close attention on providing specifics and justifications where its been requested by the reviewers. Please go through the itemized list of comments that have been provided by the reviewers to amend your paper, and provide a response as you deem appropriate. Reviewer 2 has recommended major revisions, and I expect the authors will work on those in addition to some minor clarifications that have been suggested.

Reviewers' comments:

Reviewer's Responses to Questions

**Comments to the Author**

1. Is the manuscript technically sound, and do the data support the conclusions?

Reviewer #1: Yes

Reviewer #2: Partly

Reviewer #3: Yes

2. Has the statistical analysis been performed appropriately and rigorously? 

Reviewer #1: Yes

Reviewer #2: N/A

Reviewer #3: Yes

3. Have the authors made all data underlying the findings in their manuscript fully available?

Reviewer #1: Yes

Reviewer #2: No

Reviewer #3: Yes

4. Is the manuscript presented in an intelligible fashion and written in standard English?

Reviewer #1: Yes

Reviewer #2: Yes

Reviewer #3: Yes

5. Review Comments to the Author

Reviewer #1: This is a great paper and one that is of high value. The authors should be commended for their work. And as a reviewer, thanks for taking the time writing this to a high quality prior to submitting. I have a few minor things that should be addressed prior to publication:

1. Maps are static snapshots of data – but we know water is very dynamic, especially in the upper layers. The authors discuss the relationship between precip and groundwater in the upper layers and do make mention to the stability of their estimates, but do not fully discuss how these maps may change through time. Can we expect the upper layer maps to be static in ways similar to the lower layers?

2. The maps are missing scales and N arrows. Outlines of aquifer regions can be better displayed.

3. I am curious why 25km cells were used to visualize the maps. With so many data points, it seems a smaller cell size would be possible. Was this done to minimize the gray area (i.e., places with no/missing data)?

4. These types of big data papers are becoming more common but are still new to many readers – so having better documentation on the actual technology used in the data processing will help improve this new field, as well as acclimate others to the tools that are out there. You mention R packages, but in your acknowledgements, you also talk about high performance computing and data storage. Based on the data in this study, I am assuming there are other software programs used to help manage the data, as well as GIS to visualize the data. What were the technical specs on the computer used in this analysis (e.g., RAM). This should all be included in the methods, and will help statements this like “Because the computation time required to generate 3-d semivariograms from the full dataset would have been prohibitive” make sense.

Reviewer #2: This study tried to give a picture of the national-scale groundwater resources by compiling and analyzing existing groundwater data. Some of the findings are interesting, but many parts of the method section need much more and better descriptions and justification. Many parts of the result section read like pure speculations rather than being supported by the results. The goal of this study is still not clear, and it would be great if the authors can provide examples of showing how the analysis results and findings can help understand groundwater systems in practice. The descriptions and interpretation of the results are a bit scattered; I think this study has a couple of interesting and meaningful findings that the authors want to highlight and further describe in the manuscript. I think splitting the result section into result and discussion sections would help improve the flow and readability. Finally, the authors may want to select a couple of states to explain and interpret the results in detail. Here are my specific comments.

Line 61: Is this the same as what mentioned in Line 46? This looks like important, as it is one of the sources of data used in this study. Please provide a description about the CONUS somewhere in the manuscript.

Lines 71 to 72: Please describe the “state-level geodatabases” and “U.S. Geological Survey where state-agency data were unavailable” in terms of the data spatial coverage and temporal/spatial resolutions. I wonder if the spatial/temporal resolutions of the data compiled from the different sources (or states) are consistent with each other. If not, please explain who the inconsistency was considered in the analysis and prediction.

Line 82: Is the number of samples, 210, large enough for this national scale analysis? Please justify the selection of the samples.

Lines 92 to 93: Please describe the sources of the information about the origin of tap water.

Line 100: Please provide an overview of the well and groundwater isotope databases used to develop a gridded, 3-d map of aquifer in this study, in terms of the quantity and quality.

Line 108: Please justify the selection the spatial resolution, 25 km X 25 km, of a regular grid used in this study (rather than others such as 10 km X 10 km or 100 km X 100 km). I believe the cell size or spatial resolutions are critical to this study, as the determine the accuracy as well as precision of the final results or prediction. Please discuss the sensitivity of the analysis or prediction results to the cell sizes somewhere in the manuscript.

Lines 117 to 118: Please provide technical details of the variogram analysis such as parameter values used in the spatial interpolation (or extrapolation).

Lines 124 to 126: It is still not clear why independent semivariogram modeling and kriging were used in this study. If the 3-d variograms showed systematic autocorrelation, I wonder what alternative analysis methods would be used.

Lines 131 to 134: The consistency of the isotope measurements would not be maintained by the averaging processes taken by the layers. I am not sure the “independence” assumption would be valid for the analysis.

Line 136: What does “modern” mean here?

Lines 136 to 137: I am not sure what “local groundwater – prediction isotope differences” mean or how they can be useful. In the result section, I found many discussions about this, but I am not sure the discussions are supported by the results.

Lines 140 to 141: I am not sure the tap water samples can be a good dataset for the model validation, as tap water can travel longer than 2.5 km (the cell size or spatial resolution of the analysis or prediction) from its source (groundwater).

Lines 154 to 164: I believe this paragraph may belong to the method sand data section.

Lines 191 to 193: Please clarify how such a dataset (or “product”) can be useful in practice.

Lines 189 to 213: Please describe the analysis results and compare them by states in a quantitative manner, such as using some descriptive statistics.

Lines 220 to 225: These finds are interesting. Please further describe these using plots that can show the close associations between them.

Lines 225 to 229: These are also interesting, but the authors may want to cite studies that support these finding. In addition, I think the analysis methods used in this study might have affect the results.

Reviewer #3: The authors present a new approach to model 3-D groundwater isoscapes on a large scale with an impressive compilation of groundwater isotope data across the United States. They applied both 2-dimensional and 3-dimensional variogram analysis and kriging to generate the first 3-D groundwater isoscape. The paper is well-written with appropriate literature background and context. The figures are neat and informative. I’m confident that the manuscript, the isotopic data, and the 3-D isoscape products will be useful to scientists interested in both basic areas such as hydrology and ecology, but also applied areas such as forensic sciences. The authors did an excellent job explaining possible drivers of groundwater's δ18O spatial pattern, both in-depth and in 2 dimensions, as well as demonstrating the relationship of groundwater's δ18O to that of meteoric and tap water.

I believe the paper would benefit from a process-based model involving empirical relationships with the possible drivers of the spatial and depth pattern, especially considering the need to extrapolate the information to regions with few samples. However, it is worth noting that the authors incorporated the complexity of 3-D groundwater stable isotope mapping and opened up a whole new range of possibilities for refining and using the isoscapes presented in the paper. Additionally, the authors are aware and pointed out the models' limitations, indicating ways to overcome these limitations in future works. With all that said, I do recommend this paper for publication in Plos One.

Minor comments:

Line 88: In δ = (Rsample – Rstandard) / Rstandard, it remained to show that the δ value is multiplied by a thousand (δ = (Rsample – Rstandard) / Rstandard * 1000).

Fig 2: The polygons showing the extent of the major aquifers could be more evident. Some readers may have difficulty seeing it.

Line 192: The word “that” is written twice.

Line 313 – 315: The authors say the error of precipitation isoscapes may be one of the four factors that can contribute to observed groundwater-precipitation offsets. I wondered what the behavior of observed groundwater values against the regionalized cluster-based water isotope prediction (RCWIP) might look like (Terzer et al. 2013), or even the more recent RCWIP2 (Terzer-Wassmuth et al. 2021).

Fig S1; S5; S6; S7: I cannot see the cubeView of the supplementary figures. I can open the figures, but only a gray cube with the 2-D map appears on the cube surface. If the intention was to show the data in 3-D, there had to be an error because I can't see it. If the intention was to show in 2-D over a 3-D cube, I suggest presenting it as a conventional map like the other figures in the paper.

6. PLOS authors have the option to publish the peer review history of their article (what does this mean?). If published, this will include your full peer review and any attached files.

Reviewer #1: No

Reviewer #2: No

Reviewer #3: **Yes: **João Paulo Sena-Souza

---

## [Decision Letter · Decision Letter 1]

26 Nov 2021

PONE-D-21-21283R1A 3-D groundwater isoscape of the contiguous USA for forensic and water resource sciencePLOS ONE

Dear Dr. Bowen,

Thank you for submitting your manuscript to PLOS ONE. After careful consideration, we feel that it has merit but does not fully meet PLOS ONE’s publication criteria as it currently stands. Therefore, we invite you to submit a revised version of the manuscript that addresses the points raised during the review process.

Dear Dr. Bowen. Thank you for providing your revised manuscript. Based on the revisions you have provided we believe the manuscript is worthy of publication in PLOS ONE. However, a few minor comments will need to be clarified before it can be accepted. In particular Reviewer 2 has itemized 7 points that will need clarifications to be incorporated as you provide your revision.

sincerely,

Dr. Bhadha.

(ACADEMIC EDITOR)

We look forward to receiving your revised manuscript.

Kind regards,

Jehangir H. Bhadha, Ph.D.

Academic Editor

PLOS ONE

Journal Requirements:

Reviewers' comments:

Reviewer's Responses to Questions

**Comments to the Author**

1. If the authors have adequately addressed your comments raised in a previous round of review and you feel that this manuscript is now acceptable for publication, you may indicate that here to bypass the “Comments to the Author” section, enter your conflict of interest statement in the “Confidential to Editor” section, and submit your "Accept" recommendation.

Reviewer #1: All comments have been addressed

Reviewer #2: All comments have been addressed

Reviewer #3: (No Response)

2. Is the manuscript technically sound, and do the data support the conclusions?

Reviewer #1: Yes

Reviewer #2: Partly

Reviewer #3: (No Response)

3. Has the statistical analysis been performed appropriately and rigorously? 

Reviewer #1: Yes

Reviewer #2: I Don't Know

Reviewer #3: (No Response)

4. Have the authors made all data underlying the findings in their manuscript fully available?

Reviewer #1: (No Response)

Reviewer #2: No

Reviewer #3: (No Response)

5. Is the manuscript presented in an intelligible fashion and written in standard English?

Reviewer #1: Yes

Reviewer #2: Yes

Reviewer #3: (No Response)

6. Review Comments to the Author

Reviewer #1: (No Response)

Reviewer #2: 1. “Line 61 (original ms line numbers) is in the final paragraph of the introduction, and states that our work begins with the development of a 3-d model for aquifer presence. Line 46 is in a paragraph describing previous work and references a study that used machine learning to model shallow groundwater isotope ratios. The two statements are unrelated except that the term CONUS appears in both (in both the previous study and ours this is the spatial domain under consideration). We have reviewed the relevant text and believe that the meaning is clear, including the distinction between the prior study and the introduction of our own work, given the context in which these statements appear. The term CONUS is defined on line 46. The datasets used in our work are described in the subsequent section (Materials and Methods).”:

The expression is still not clear if CONUS means “contiguous United States” or the name of a dataset used in the study of (28). I think the authors may want to separate (CONUS) from (28) or [28]; it may be helpful if the expression can be revised to “conterminous United States (or CONUS) [28].”

2. “We are confused by this request, since the subsection “Materials and Methods, Data” which immediately proceeds the text referenced by the reviewer provides the requested information. We assume there’s just some confusion about the fact that the data sources described in that section are in fact those which are used in the aquifer mapping and have attempted to clarify this by editing the referenced text to read “We used the well and groundwater isotope databases described in the previous section…”:

I am sorry that the authors were confused with this suggestion. What I would like to suggest was to provide “an overview” of the data used as clearly stated in the comment. I know the authors described the data in the previous section, but it was hard for me to clearly understand the overall picture of data used in terms of their quantity and quality, as many different types (well depths, isotope ratios, and tap water samples) of data that came from different sources (as describe in Tables S1 and S2) were used for different purposes (model development and validation). For example, it may be helpful to have a table classifying the datasets (with abbreviations or simple names) by purposes, sources, and types.

3. “These data represent a small fraction of the total dataset. We have added a statement here that clarifies that these data were collected to fill “identified gaps in the compiled dataset”:

Please elaborate on the “identified gaps.” I also found that the spatial coverages of data used in this study were not clearly described in the manuscript; please try to map the coverages (using points and/or polygons) of the individual data.

4. “The text referenced here states that “…the 3-d variograms showed no systematic autocorrelation of isotope δ values in the depth dimension…”, and subsequent text in the Results/Discussion section goes on to discuss this result in more detail. Based on the reviewer’s comment is seems they may have mis-read this statement. The lack of autocorrelation in the depth domain reported here provides justification for proceeding with the 2-d modeling.:

I believe I did read the statement correctly. My question was “if the 3-d variograms showed systematic autocorrelation, I wonder what alternative analysis methods would be used” as the authors understood. I was curious about alternative methods that the authors might use when they found there was systematic autocorrelation of isotope δ values. The authors did not know if 3-d variograms might show so systematic autocorrelation or strong systematic autocorrelation, and I was wondering what kinds of methods and approaches we might be able to use if there was STRONG systematic autocorrelation.

5. “All semivariagram parameters appear in Table 1 in the manuscript and are discussed in the Results/Discussion section.”:

Please mention Table 1 at the end of this description.

6. Regarding the first paragraph of the “Results and Discussion” section:

It must be up to the authors, but I still believe this belong to the method and data section, as this is a summary of raw data compiled from the sources (rather than being analyzed or processed).

7. Regarding the relationship between groundwater and precipitation isotopes:

I believe a plot similar to Figure 6 (of the original manuscript) can be helpful.

Reviewer #3: (No Response)

7. PLOS authors have the option to publish the peer review history of their article (what does this mean?). If published, this will include your full peer review and any attached files.

Reviewer #1: No

Reviewer #2: No

Reviewer #3: **Yes: **João Paulo Sena-Souza

---

## [Editor Report · Decision Letter 2]

9 Dec 2021

A 3-D groundwater isoscape of the contiguous USA for forensic and water resource science

PONE-D-21-21283R2

Dear Dr. Bowen,

We’re pleased to inform you that your manuscript has been judged scientifically suitable for publication and will be formally accepted for publication once it meets all outstanding technical requirements.

Kind regards,

Jehangir H. Bhadha, Ph.D.

Academic Editor

PLOS ONE

Additional Editor Comments (optional):

Dear authors. Thank you for providing the edits. After reviewing the second round of edits, I am fully satisfied and have no hesitation in accepting the paper for publication.
---

## [Editor Report · Acceptance letter]

13 Dec 2021

PONE-D-21-21283R2 

A 3-D groundwater isoscape of the contiguous USA for forensic and water resource science 

Dear Dr. Bowen:

I'm pleased to inform you that your manuscript has been deemed suitable for publication in PLOS ONE. Congratulations! Your manuscript is now with our production department. 

Kind regards, 

on behalf of

Dr. Jehangir H. Bhadha 

Academic Editor

PLOS ONE